# Antibacterial efficacy and possible mechanism of action of 2-hydroxyisocaproic acid (HICA)

**Amila S. N. W. Pahalagedara**[1,2¤], **Steve Flint**[2], **Jon Palmer**[2], **Gale Brightwell**[1,3], **Tanushree B. Gupta**[1]*

**1** Food System Integrity Team, AgResearch Ltd, Hopkirk Research Institute, Massey University, Palmerston North, New Zealand, **2** School of Food and Advanced Technology, Massey University, Palmerston North, New Zealand, **3** New Zealand Food Safety Science and Research Centre, Massey University, Palmerston North, New Zealand

¤ Current address: Department of Food Science, Aarhus University, Aarhus, Denmark
* tanushree.gupta@agresearch.co.nz

**Data Availability Statement:** All relevant data are within the paper and its Supporting Information files as Supplementary data.

## Abstract

The exploitation of natural antimicrobial compounds that can be used in food preservation has been fast tracked by the development of antimicrobial resistance to existing antimicrobials and the increasing consumer demand for natural food preservatives. 2-hydroxyisocaproic acid (HICA) is a natural compound produced through the leucine degradation pathway and is produced in humans and by certain microorganisms such as lactic acid bacteria and *Clostridium* species. The present study investigated the antibacterial efficacy of HICA against some important bacteria associated with food quality and safety and provided some insights into its possible antimicrobial mechanisms against bacteria. The results revealed that HICA was effective in inhibiting the growth of tested Gram-positive and Gram-negative bacteria including a multi-drug resistant *P. aeruginosa* strain in this study. The underlying mechanism was investigated by measuring the cell membrane integrity, membrane permeability, membrane depolarisation, and morphological and ultrastructural changes after HICA treatment in bacterial cells. The evidence supports that HICA exerts its activity via penetration of the bacterial cell membranes, thereby causing depolarisation, rupture of membranes, subsequent leakage of cellular contents and cell death. The current study suggests that HICA has potential to be used as an antibacterial agent against food spoilage and food-borne pathogenic bacteria, targeting the bacterial cell envelope.

## Introduction

The microbial quality and safety of food products during processing, storage, and until the consumption, is maintained by controlling the growth of food spoilage and food-borne pathogens using various food preservation techniques including the use of antimicrobial compounds. In food preservation, the use of natural antimicrobial compounds has been favoured over synthetic ones, due to consumer awareness and concerns over adverse health effects of some synthetic chemicals [1]. However, synthetic preservatives are still widely used in the food industry; out of 34 antimicrobials approved by the European Food Safety Authority, 32 are

**Funding:** This work was funded by the Strategic Science Investment Fund (SSIF), AgResearch Ltd., New Zealand. The funders had no role in study design, data collection and analysis, decision to publish, or preparation of the manuscript.

**Competing interests:** The authors have declared that no competing interests exist.

synthetic agents, and only 3 (nisin, lysozyme and natamycin) are natural [2]. The effectiveness of existing food antimicrobials has been challenged by the development of antimicrobial resistant bacterial strains. For instance, nisin, the most widely used FDA approved bacteriocin has lost its efficacy due to the development of nisin resistance in bacteria [3, 4]. Therefore, there is a great demand for novel natural antimicrobials that can be safely applied in food preservation.

2-hydroxyisocaproic acid (HICA), which is also referred as 2-hydroxy-4-methylvaleric acid, DL-leucic acid, 2-hydroxy-4-methylpentanoic acid and alpha-hydroxyisocaproic acid, is a leucine metabolite with hydroxy substituent at the 2-position and a methyl substituent at the 4-position [5]. This compound is produced as a by-product of the leucine degradation pathway in humans and certain microorganisms. In humans, it is produced in tissues including connective tissues and muscles [6] and regarded as a physiological agent present in the human body at low concentrations. The healthy adult usually contains around 0.1–0.25 mmol/L HICA in their plasma. Plasma HICA level increases after exercise or during prolonged fasting due to the breakdown of proteins for energy [7]. HICA has also been reported to be a microbial metabolite produced during fermentation of animal proteins [8]. Mainly lactic acid bacteria (LAB) such as *Lactobacillus plantarum*, *Lactococcus lactis*, *Lactobacillus plantarum*, *Lactobacillus brevis*, and *Leuconostoc mesenteroides* have been reported to synthesise HICA. Other microorganisms reported to produce HICA are *Clostridium difficile* and *Peptostreptococcus anaerobicus* [9–15]. Fermented food products such as certain cheeses, soy sauce, wines, and kimchi have been found to contain HICA [15, 16]. Our previous study putatively identified HICA from *Clostridium* grown conditioned/spent medium, which exhibited promising antibacterial activity [17].

HICA has previously shown to possess antimicrobial activity against clinically important bacteria and fungi [18, 19]. However, its potential application in food preservation through inhibiting the growth of food spoilage and food-borne pathogenic bacteria has not been investigated to date. We suggest that HICA could be used as a novel and safe antimicrobial compound in food preservation as it is a natural compound in food products and is metabolised by the human body. Therefore, the aim of the present study was to evaluate the antibacterial efficacy of HICA against different bacteria associated with food quality and safety and provide some insights into its possible antibacterial mechanism. Antibacterial efficacy of HICA was evaluated by determining minimum inhibitory concentration (MIC) and minimum bactericidal concentration (MBC) against some Gram-positive and Gram-negative bacteria. The underlying mechanism was investigated by measuring the cell membrane integrity, membrane permeability, membrane depolarisation, and morphological and ultrastructural changes after HICA treatment in bacterial cells.

## Materials and methods

### Bacteria and growth conditions

*Bacillus mycoides* ATCC6462, *Bacillus cereus* NZRM5, *Pseudomonas aeruginosa* ATCC25668, *Pseudomonas aeruginosa* NZRM4034, and *Staphylococcus aureus* NZRM917 were purchased from Environmental Science and Research (ESR), New Zealand. *Escherichia coli* O157:H7 NCTC12900 was obtained from the National Collection of Type Cultures (NCTC), Public Health England. *Bacillus cereus* M4 and *Escherichia coli* AGR3789 are milk and soil isolates, respectively. *Shewanella putrefaciens* SM 26 and *Serratia proteamaculans* ENT 68 are meat isolates. *Bacillus subtilis* F2MCUH1, *Paenibacillus odorifer* F1OSP28, *Pseudomonas lundensis* F2MCUH2, and *Pseudomonas fragi* F1NBUH38 are farm environmental isolates. All bacteria except reference strains used in this study, were obtained from our laboratory culture

collection (Food System Integrity Team, AgResearch Ltd., New Zealand). All bacteria were first revived on Sheep blood agar (SBA) plates (Fort Richard Laboratories, New Zealand) from frozen glycerol stocks. A single colony of each bacterium on the SBA plate was inoculated into Muller-Hinton broth (MHB) (Fort Richard Laboratories, New Zealand) and incubated overnight at 35˚C, except *Shewanella putrefaciens* SM26, *Serratia proteamaculans* ENT68, *Paenibacillus odorifer* F1OSP28, *Pseudomonas lundensis* F2MCUH2, and *Pseudomonas fragi* F1NBUH38 which were incubated at 25˚C. Broth cultures were used for various assays in this study.

## Preparation of DL-2-hydroxyisocaproic acid (HICA) stock solution

A 100 mg/mL stock solution of HICA was prepared by dissolving DL-HICA (Sigma-Aldrich, USA) in sterile Milli Q water (Milli-Q®, Germany) and filter sterilised using a 0.22 μm membrane filter (Millipore, Ireland). The solution was further diluted to different concentrations as required in each assay.

## Determination of minimum inhibitory concentrations (MICs) and minimum bactericidal concentration (MBCs) of HICA

Minimum inhibitory concentration (MIC) and minimum bactericidal concentration (MBC) of HICA against various bacteria were determined by the microplate turbidimetric assay method as described earlier [17, 20] with some modifications. An overnight bacterial culture grown in MHB was diluted to achieve a final inoculum of $5 \times 10^5$ CFU/mL. A working two-fold dilution series (0.5, 1, 2, 4, 8, 16, 32, 64 mg/mL) of HICA (Sigma-Aldrich, USA) was prepared and added to respective wells to achieve final assay concentrations ranging from 0.25–32 mg/mL. Bacterial culture (50 μL) was added to MHB (50 μL) and a two-fold dilution series of HICA (100 μL) or sterile water (untreated control) in a 96-well flat bottom microtiter plate (Thermo scientific, Denmark) and covered with a Breathe-Easy® sealing membrane. The plate was incubated in a spectrophotometer at 35˚C or 25˚C depending on the optimum growth temperature of bacteria. The pH of the final suspension (HICA + media + bacteria) ranged from 7 (0.25 mg/mL) to 3 (32 mg/mL). The bacterial growth was determined by measuring the optical density (OD) at 595 nm for 24 h. MIC was the lowest concentration of HICA at which there was no growth of test bacteria (less than 0.01 $OD_{595nm}$ increase during 24 h incubation). The MBC was assessed by removing the media from each well after 24 h assay and sub-culturing them on SBA plates. SBA allows maximum recovery of bacteria after exposing to HICA including bacteria with sub-lethal injuries. MBC was the minimum concentration of HICA required to completely kill test bacteria (no colonies on SBA plates). The experiments were conducted in triplicates on different occasions and used to assess the MIC and MBC against all test bacteria.

## Bacterial cell viability assay for both Gram-positive and Gram-negative bacteria

The overnight bacterial cultures of *B. cereus* NZRM5, *B. cereus* M4, *S. aureus* NZRM917, *E. coli* O157:H7 NCTC 1200, *E. coli* AGR3789, *P. aeruginosa* NZRM981, and *P. aeruginosa* NZRM4034 were grown in MHB and adjusted to achieve a cell density of approximately $1 \times 10^7$ CFU/mL. HICA (4 mg/mL, pH 5.6) or sterile water (untreated control) was added to the bacterial culture and incubated at 35˚C for different time intervals (0–180 min). The bacterial cell viability was determined using the BacTiter-Glo microbial cell viability assay kit (Promega, USA) and following manufacturer's instruction at various treatment times. Briefly,

BacTiter-Glo reagent was prepared by combining lyophilised BacTiter-Glo™ enzyme/substrate mixture with the buffer at room temperature. One hundred microlitres of HICA treated/untreated (sterile water) bacterial culture was transferred to an opaque-walled 96 well plate and combined with 100 μL of BacTiter-Glo reagent. After mixing the contents in the dark for 5 min, the luminescence intensity was measured using a Varioskan™ LUX multimode microplate reader (Thermo Fisher, USA). The relative cell viability at various treatment times was calculated as the percentage of untreated cells (untreated control). The experiments were conducted in triplicates on different occasions and used to assess the bacterial cell viability.

## Evaluation of the loss of cell membrane integrity of Gram-positive and Gram-negative bacteria

Bacterial cell membrane integrity was assessed by CellTox™ green cytotoxicity assay kit (Promega, USA) according to manufacturer's instruction. Briefly, the overnight bacterial cultures of *B. cereus* NZRM5, *B. cereus* M4, *S. aureus* NZRM917, *E. coli* O157:H7 NCTC 1200, *E. coli* AGR3789, *P. aeruginosa* NZRM981, and *P. aeruginosa* NZRM4034 grown in MHB were harvested by centrifugation at $10,000 \times g$ for 5 min. The cell pellets were re-suspended in MHB to achieve $1 \times 10^7$ CFU/mL. HICA (4 mg/mL) or sterile water was added to the bacterial cultures and incubated at 35°C for different time intervals (15–120 min). The cells treated with sterile water served as the untreated control. 2X CellTox™ green reagent was prepared by combining 30 μL of CellTox™ green dye with 15 mL of assay buffer. One hundred micro-litres of the sample were transferred to an opaque-walled 96 well plate at different treatment times and combined with 100 μL of 2X CellTox™ green reagent. The plates were incubated in the dark for 15 min at room temperature. The fluorescence intensities of bacterial cultures were measured at 490 nm excitation and 520 nm emission wavelengths using a Varioskan™ LUX multimode microplate reader.

## Outer membrane permeability of Gram-negative bacteria

The effect of HICA on the permeability of outer membrane of Gram-negative bacteria was investigated using the non-polar fluorescence probe 1-N-phenylnaphthylamine (NPN) as described previously [21] with minor modifications. Briefly, the overnight bacterial cultures of *E. coli* O157:H7 NCTC 1200, *E. coli* AGR3789, *P. aeruginosa* NZRM981, and *P. aeruginosa* NZRM4034 grown in MHB were diluted to achieve 0.5 $OD_{595nm}$. The cells were harvested by centrifugation at 10,000 x g for 5 min, washed twice with assay buffer (HEPES, pH 7.2) and resuspended in half of the original volume with the assay buffer. For the treatment groups, HICA (4 mg/mL) was added in an opaque-walled 96 well plate and combined with 40 μM NPN (50 μL) (Sigma-Aldrich, USA). Bacterial cell suspension (50 μL) was added to the wells immediately before the fluorescence measurements. For the untreated control group, HICA was replaced by the assay buffer and the background control was assay buffer with NPN. The fluorescence intensities were measured at an excitation wavelength of 350 nm and emission wavelength of 420 nm for 10 min at 30 s intervals using a Varioskan™ LUX multimode microplate reader. The NPN uptake was calculated using the formula below (1) and the values obtained for 10 min were averaged to get the normalised fluorescence intensity. $F_{obs}$ is the observed fluorescence intensity after HICA treatment, $F_{con}$ is the fluorescence intensity of NPN with bacterial cells and $F_B$ is the fluorescence intensity of NPN without bacterial cells in the presence of assay buffer.(1)

$$NPN\ uptake = (F_{obs} - F_B) - (F_{con} - F_B) \tag{1}$$

## Cytoplasmic membrane depolarisation assay

The cytoplasmic membrane depolarisation activity of HICA was determined using the voltage sensitive fluorescence dye 3,3′-Dipropylthiadicarbocyanine iodide [DiSC$_3$(5)] (Sigma-Aldrich, Czech Republic) as described previously [22] with minor modifications. *B. cereus* NZRM5, *B. cereus* M4, *S. aureus* NZRM917, *E. coli* O157:H7 NCTC 1200, *E. coli* AGR3789, *P. aeruginosa* NZRM981, and *P. aeruginosa* NZRM4034 were grown overnight in MHB at 35˚C and diluted to 0.3 OD$_{595}$ with fresh MHB supplemented with 0.5 mg/mL Bovine Serum Albumin (BSA) (Sigma-Aldrich, USA). The diluted bacterial cell suspension (130 μL) was transferred to an opaque-walled 96 well plate and combined with 20 μL of 60 μM DiSC$_3$(5) (Sigma-Aldrich, Czech Republic). The fluorescence quenching was then measured using the Varioskan™ LUX multimode microplate reader at an excitation wavelength of 610 nm and emission wavelength of 660 nm, until a stable fluorescence signal was achieved. Following the addition of HICA (4 mg/mL) or sterile water to corresponding wells, fluorescence was immediately measured and monitored for 25 min at 60 s intervals with 10 s of vigorous shaking before each measurement. Samples with added sterile water served as the untreated controls and used to compare the change in the fluorescence intensity of HICA-treated group.

## Scanning electron microscopy (SEM)

*B. cereus* NZRM5 and *P. aeruginosa* ATCC25668 were grown overnight in MHB and diluted with fresh MHB to achieve $1 \times 10^7$ CFU/mL. Bacterial cultures were then treated with 4 mg/mL HICA, or sterile water followed by incubation at 35˚C for a total of 120 min. Samples were drawn at various treatment time intervals and the cells were harvested by centrifugation at 10,000 x g for 10 min, washed twice with 0.1 M phosphate-buffered saline (PBS) and used for SEM technique. The cells treated with sterile water served as the untreated control for comparing morphological changes after HICA treatment.

The primary fixation of cells was done with 0.1 M PBS containing 3% glutaraldehyde (Merck, Germany) for at least 8 h. The cell suspensions were passed through 0.4 μm Isopore™ membrane filters (Millipore, Ireland) to place the cells on the membrane. The membranes were washed three times with 0.1 M PBS for 15 min each. Dehydration was performed with rising ethanol concentrations as follows; 25%, 50%, 75%, 95%, and 100% for 15 min each and a final 100% for 1 h. All the samples were dried using a critical point drying apparatus (Quorum technologies, UK), mounted on the aluminium stubs and sputter coated with approximately 100 nm thickness of gold (Bal-Tec, USA). Bacteria were imaged using the FEI Quanta 200 scanning electron microscope (ThermoFisher, USA) at an accelerating voltage of 15 kV.

## Transmission electron microscopy (TEM)

The cell treatments and the initial sample preparation was carried out using the same procedure described under SEM technique. The primary fixation of cells was done in 0.1 M PBS containing 3% glutaraldehyde. The fixed cells were injected into 3% agarose tubes to make enclosed capsules and stored with 3% glutaraldehyde in 0.1 M sodium cacodylate buffer (pH 7.2) for at least 24 h. Capsules were subsequently washed with 0.1 M sodium cacodylate buffer, followed by post fixation with 1% osmium tetroxide in 0.1 M sodium cacodylate buffer. After washing with 0.1M sodium cacodylate buffer, samples were dehydrated with rising acetone concentrations as follows: 25%, 50%, 75%, 95%, 100%, 100%, 100% for 45 min each. Samples were then incubated in a 1:1 mixture of resin and acetone overnight and then with fresh 100% resin (812 resin, ProSciTech, Australia) for another 8h. This step was repeated for four more times. Samples were then embedded in moulds with fresh resins, cut into one-micron thin layers using an ultramicrotome (Leica, Germany) and heat fixed onto glass slides. Ultrathin sections (70 nm) were cut using a diamond

knife (Diatome, Switzerland), stretched with chloroform, and mounted on copper grids. Copper grids were stained with saturated uranyl acetate, washed with 50% ethanol and MilliQ water followed by staining with lead citrate and washing with MilliQ water. Bacteria were imaged using FEI Tecnai™ G$^2$ Spirit BioTWIN (Czech Republic) transmission electron microscope using a side mounted TEM CCD camera (Olympus, Germany).

## Statistical analysis

Three independent replicates of the experiments were carried out. The single factor Analysis of Variance (ANOVA) was performed to check the statistical significance between the untreated control and various treatment groups. The values with $P < 0.05$ were considered statistically significant.

## Results

### Antibacterial efficacy of HICA

Susceptibility testing of fourteen bacteria including both reference strains and environmental/meat isolates belonging to eleven species was carried out to investigate the antibacterial efficacy of HICA (Table 1). HICA inhibited the growth of all test bacteria in a dose-dependent manner. The antibacterial efficacy of HICA against both Gram-positive and Gram-negative bacteria used in this study was similar, showing a MIC of 1 mg/mL except for *Shewanella putrefaciens* SM26, which was the most susceptible bacteria to HICA with only 0.5 mg/mL MIC. Both multi-drug resistant *Pseudomonas aeruginosa* NZRM4034 (Ceftazidime/piperacillin resistant) and *Pseudomonas aeruginosa* ATCC25668 (clinical isolate) showed the same level of susceptibility to HICA (MIC = 1 mg/mL). According to MBC comparison, *Bacillus cereus* was found to be the most tolerant bacteria demonstrating the highest MBC value (32 mg/mL). The MBC for all other test bacteria ranged from 1 to 4 mg/mL as shown in Tables 1 and S1. The *Bacillus mycoides* reference strain and *Shewanella putrefaciens* meat isolate were the most susceptible to HICA with MBC values as low as 1 mg/mL.

### Effect of HICA on the cell viability of Gram-positive and Gram-negative bacteria

Seven bacteria comprising both reference strains and environmental isolates belonging to both Gram-positive and Gram-negative bacteria were used to assess the effect of HICA on the

**Table 1. Minimum inhibitory concentrations (MIC) and minimum bactericidal concentrations (MBC) of HICA.**

| Microorganism | MIC (mg/mL) | MBC (mg/mL) |
|---|---|---|
| *Shewanella putrefaciens* SM26 (meat isolate) | 0.5 | 1.0 |
| *Serratia proteamaculans* ENT68 (meat isolate) | 1.0 | 2.0 |
| *Escherichia coli* O157:H7 NCTC12900 | 1.0 | 2.0 |
| *Escherichia coli* AGR3789 (soil isolate) | 1.0 | 2.0 |
| *Pseudomonas aeruginosa* ATCC25668 | 1.0 | 2.0 |
| *Pseudomonas aeruginosa* NZRM4034 | 1.0 | 2.0 |
| *Pseudomonas lundensis* F2MCUH2 (environmental isolate) | 1.0 | 2.0 |
| *Pseudomonas fragi* F1NBUH38 (environmental isolate) | 1.0 | 2.0 |
| *Staphylococcus aureus* NZRM917 | 1.0 | 2.0 |
| *Bacillus mycoides* ATCC6462 | 1.0 | 1.0 |
| *Bacillus subtilis* F2MCUH1 (environmental isolate) | 1.0 | 4.0 |
| *Paenibacillus odorifer* F1OSP28 (environmental isolate) | 1.0 | 4.0 |
| *Bacillus cereus* NZRM5 | 1.0 | 32.0 |
| *Bacillus cereus* M4 (milk isolate) | 1.0 | 32.0 |

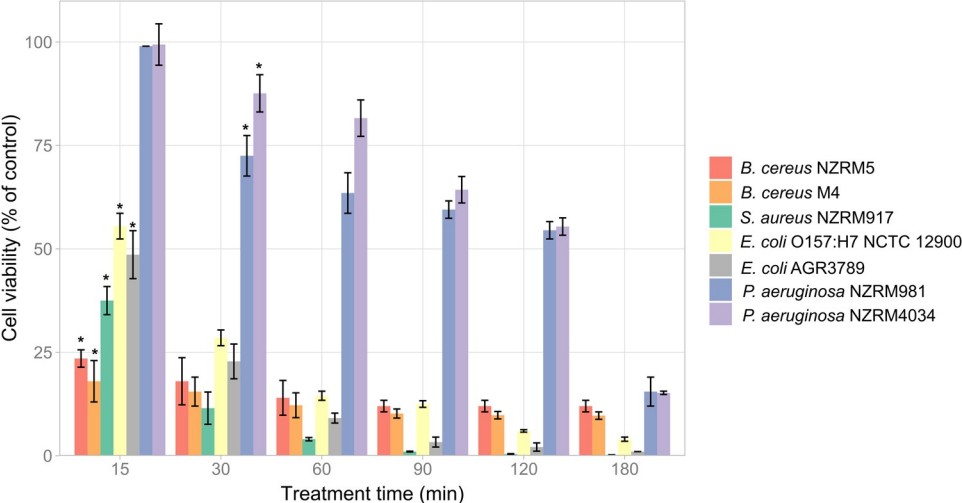

**Fig 1. Antibacterial effect of HICA on selected Gram-positive and Gram-negative bacteria.** Data are presented as mean ± (s.d.) of cell viability expressed as a percentage of untreated control of three replicates (n = 3). * Significant reduction ($P < 0.05$) compared with control.

bacterial cell viability using a luciferase bioluminescence-based method. HICA treatment reduced the cell viability of all test bacteria in a time dependant manner despite the differences in the percentage of viable cells in each bacterial group at different treatment times (Fig 1). Except *P. aeruginosa* strains, all other test bacteria displayed a significant reduction in their cell viability after 15 min of HICA treatment compared to their respective controls ($P < 0.05$). Gram-negative bacteria (*P. aeruginosa* and *E. coli*) had relatively higher viable cell percentages than Gram-positive bacteria (*B. cereus* and *S. aureus)* during the first 30 min of HICA treatment. *P. aeruginosa* strains were the most resistant to HICA showing significant reduction ($P < 0.05$) in viability after only 30 min and maintaining relatively higher percentages of viable cells compared to other test bacteria at all treatment times. Overall, HICA was shown to be effective in reducing the cell viability of tested Gram-positive and Gram-negative bacteria related to food spoilage and safety issues.

## Effect of HICA on bacterial cell membrane integrity

The effect of HICA on the bacterial cell membrane integrity was assessed using the CellTox™ green cytotoxicity assay kit. In the absence of HICA, fluorescence intensities of all bacteria were very low indicating their intact/undamaged cell membranes. After the treatment with HICA, there was a significant increase ($P < 0.05$) in the fluorescence intensity of *B. cereus* NZRM5, *B. cereus* M4 and *S. aureus* NZRM917 compared to untreated controls indicating the loss of cell membrane integrity (Fig 2). Similarly, Gram negative bacteria (*E. coli* O157:H7 NCTC12900, *E. coli* AGR3789, *P. aeruginosa* ATCC25668 and *P. aeruginosa* NZRM4034) treated with HICA lost their cell membrane integrity, indicated by a significant increase ($P < 0.05$) in fluorescence over the treatment time compared to the relevant controls (Fig 2). However, tested Gram-positive bacteria demonstrated a more rapid loss of cell membrane integrity than tested Gram-negative bacteria achieving the maximum fluorescence values within 30 min of treatment. The CellTox green dye assay showed the changes in the membrane integrity of bacteria treated with HICA compared to the untreated control of the same bacteria. The maximum fluorescence intensity levels (Relative Fluorescence Units, RFU) were different for each bacterium (Fig 2).

## Effect of HICA on the outer membrane permeability of Gram-negative bacteria

The fluorescence probe 1-*N*-phenylnapthylamine (NPN) was used to assess the activity of HICA in permeabilising the outer membranes of *E. coli* and *P. aeruginosa* strains. HICA permeabilized the outer membranes of *E. coli* O157:H7 NCTC12900, *E. coli* AGR3789, *P. aeruginosa* ATCC25668, and *P. aeruginosa* NZRM4034 in a dose dependant manner as observed by an increase in NPN fluorescence (Fig 3). Even, 0.5 mg/mL HICA (sub-MIC) showed a significant increase in the permeability of the outer membrane of all test bacteria compared to relevant untreated controls ($P < 0.05$). HICA increased the outer membrane permeability of *P. aeruginosa* NZRM4034 and *P. aeruginosa* ATCC25668 in a similar fashion.

## Effect of HICA on cytoplastic membrane potential

The effect of HICA on the cytoplasmic membrane potential of both Gram-positive and Gram-negative bacteria was assessed using the membrane potential-sensitive dye 3,3'-Dipropylthiadicarbocyanine iodide [DiSC$_3$(5)]. HICA addition induced a rapid depolarisation of the cytoplasmic membranes of both Gram-positive and Gram-negative bacteria tested in this study as evident by the rapid increase in fluorescence intensities (Fig 4). In contrast, the addition of sterile water (untreated control) showed no alteration in the cytoplasmic membrane potential of all test bacteria as measured by no rapid increase in fluorescence intensities.

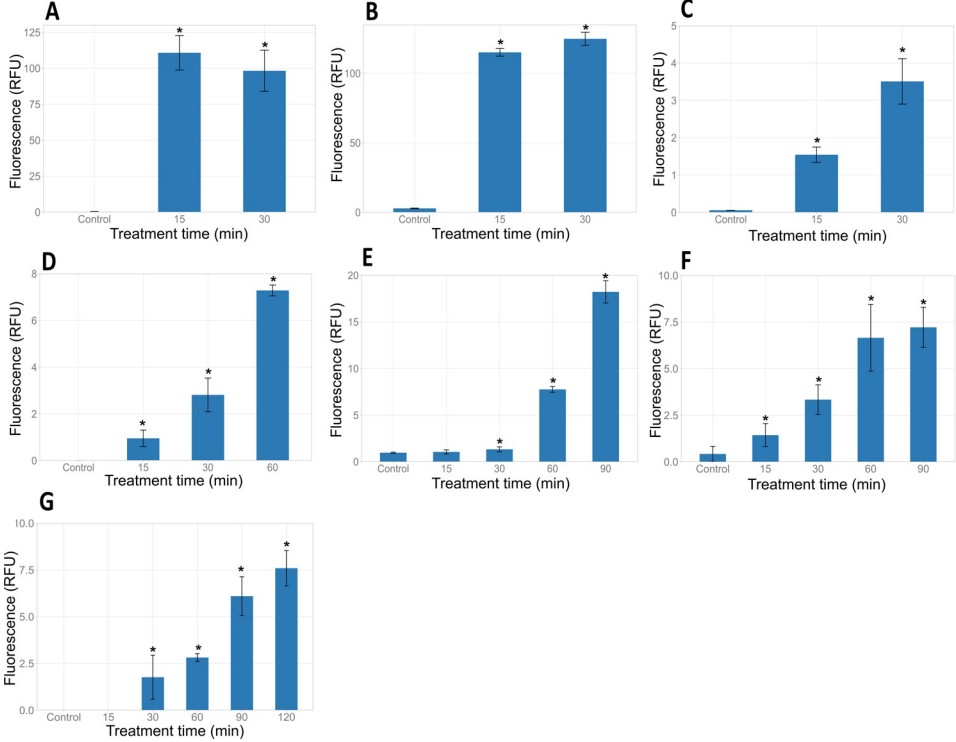

**Fig 2. Effect of HICA on the cell membrane integrity of Gram-positive and Gram-negative bacteria.** *Bacillus cereus* NZRM5 (**A**), *Bacillus cereus* M4 (**B**), *Staphylococcus aureus* NZRM917 (**C**), *Escherichia coli* O157:H7 NCTC12900 (**D**), *Escherichia coli* AGR3789 (**E**), *Pseudomonas aeruginosa* ATCC25668 (**F**) and *Pseudomonas aeruginosa* NZRM4034 (**G**) were treated with 4 mg/mL HICA, and the loss of cell membrane integrity was assessed by measuring the fluorescence intensity at various treatment times. Untreated cells were used as a control. Data are presented as mean ± (s.d.) of relative fluorescence units (RFU) of three replicates (n = 3). $^*P < 0.05$ compared with the control.

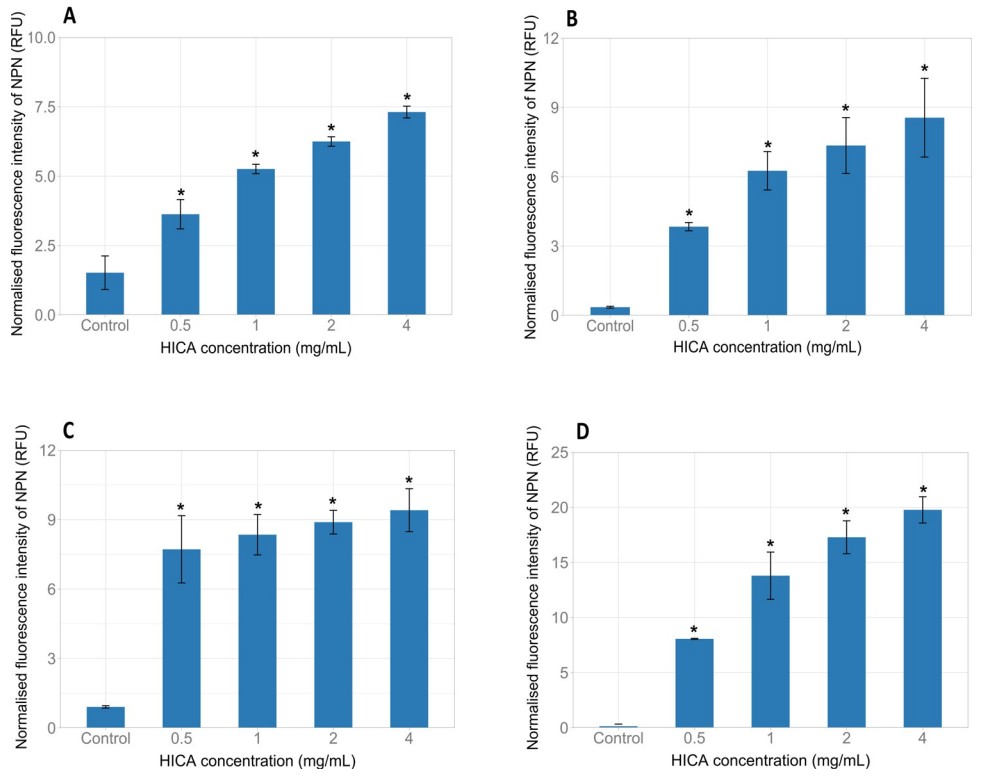

**Fig 3. Effect of HICA on the outer membrane permeability of Gram-negative bacteria.** *Escherichia coli* O157:H7 NCTC12900 (**A**), *Escherichia coli* AGR3789 (**B**), *Pseudomonas aeruginosa* ATCC25668 (**C**) and *Pseudomonas aeruginosa* NZRM4034 (**D**) were treated with various concentrations (0.5–4 mg/mL) of HICA, and the outer membrane permeability was assessed by measuring the fluorescence intensity.

## Scanning electron microscopy (SEM)

The morphological characteristics of bacterial cells treated with 4 mg/mL HICA for various time periods were observed by SEM. The untreated *B. cereus* NZRM5 cells were approximately 3–4 μm long and had a smooth and intact surface (Fig 5A). The surfaces of some cells treated with HICA for 60 min appeared corrugated and there were ruptures in these cells, however other cells looked like untreated cells (Fig 5B). The incubation of cells with HICA for 120 min caused excessive leakage of the cellular content (Fig 5C) and there were ruptures in the cell surfaces. SEM images demonstrated the distinct signs of damage to the cell envelope including roughening, rupturing and the leak of the cellular content after the HICA treatment in a time dependant manner.

In the control sample of *P. aeruginosa* ATCC25668, the cells were around 2–2.5 μm long, rod-shaped, and showed a smooth and undamaged cell structure. Additionally, there were extracellular polymeric substances (exopolysaccharide) [23] on the surfaces of untreated cells (Fig 5D). The HICA treatment for 60 min or more, reduced the size of the cells to as little as approximately 1 μm long and they lost extracellular polymeric substances on their cell surfaces. As the treatment time increased, cells became shorter (Fig 5E and 5F). However, there was no visible damage to the cell envelope and no cellular content was observed outside of HICA treated cells.

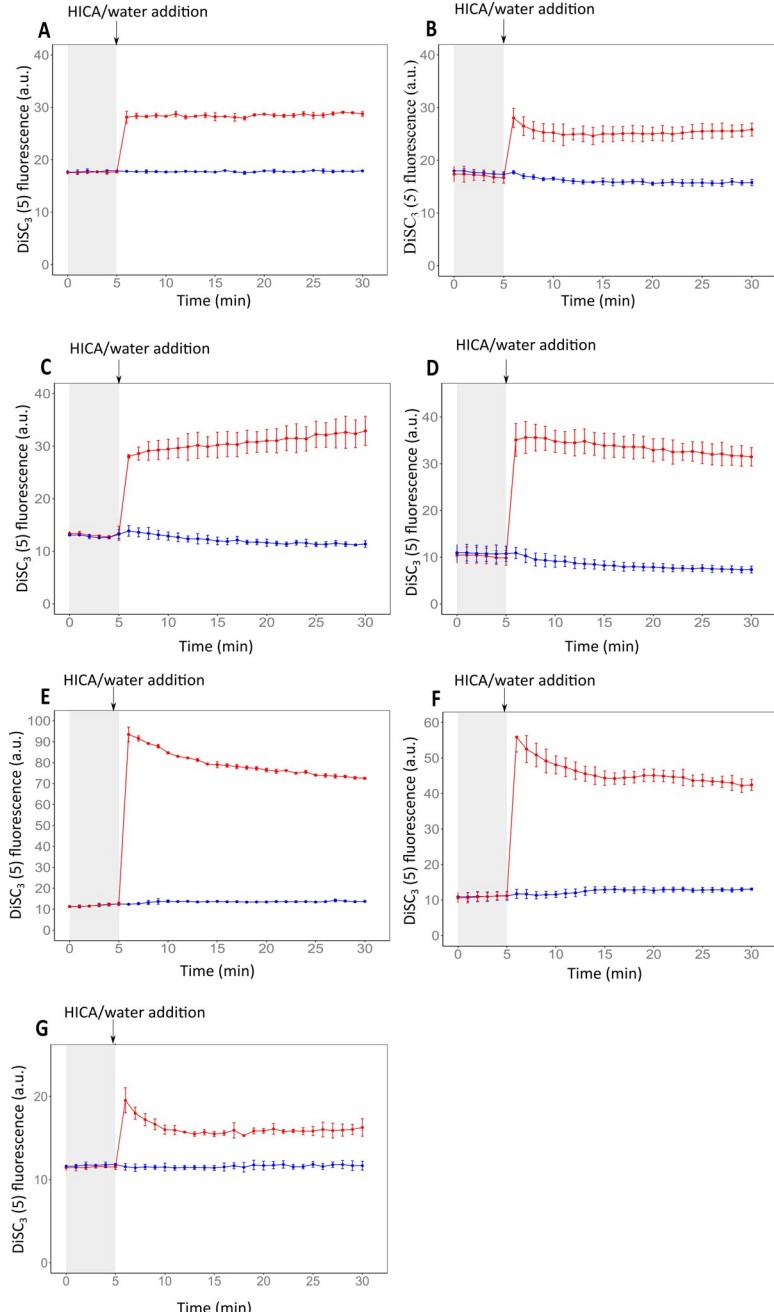

**Fig 4. Cytoplasmic membrane depolarization of Gram-negative and Gram-positive bacteria by HICA.** Membrane potential levels of *Escherichia coli* O157:H7 NCTC12900 (**A**), *Escherichia coli* AGR3789 (**B**), *Pseudomonas aeruginosa* ATCC25668 (**C**) and *Pseudomonas aeruginosa* NZRM4034 (**D**), *Bacillus cereus* NZRM5 (**E**), *Bacillus cereus* M4 (**F**), and *Staphylococcus aureus* NZRM917 (**G**) upon addition of 4 mg/mL HICA (red line) or sterile water (untreated control, blue line) were assessed by the release of the membrane potential-sensitive dye $DiSC_3(5)$ measured spectroscopically at 610 nm excitation and 660 nm emission wavelengths. The time point of HICA, or sterile water (as control) addition is highlighted with arrows. Data are presented as mean ± (s.d.) of $DiSC_3(5)$ fluorescence of three replicates (n = 3).

*B. cereus* NZRM5    *P. aeruginosa* NZRM5

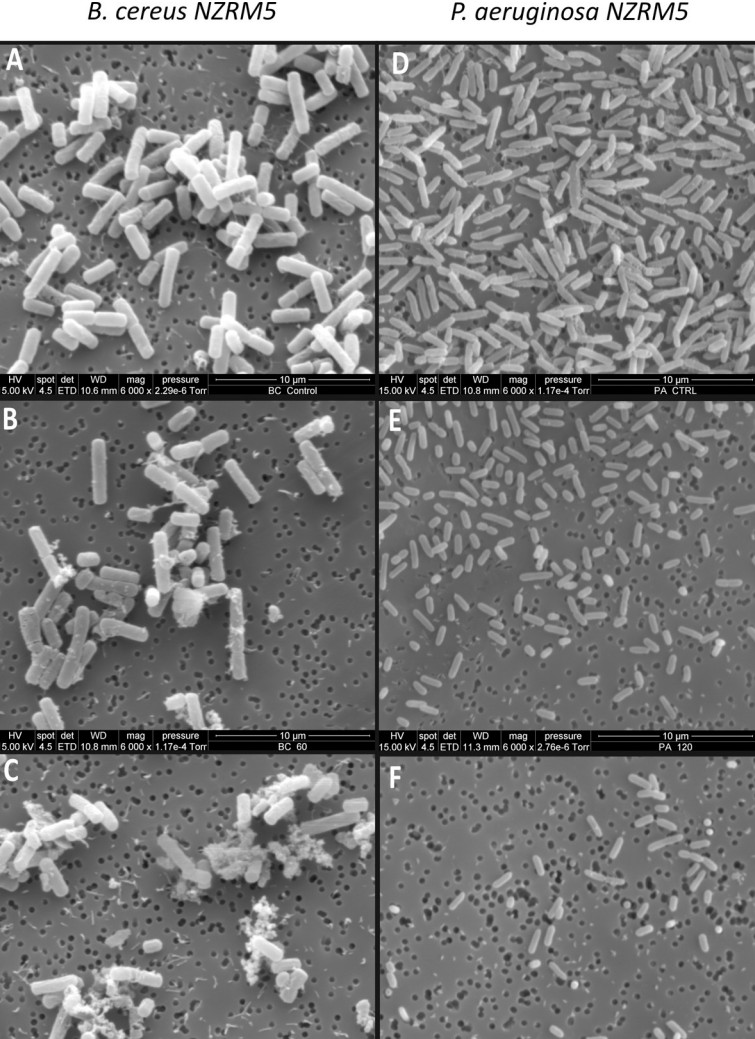

**Fig 5. Scanning electron microscopic micrographs depicting the effect of HICA on the cell morphology of *Bacillus cereus* NZRM5 and *Pseudomonas aeruginosa* ATCC25668.** The cells were treated with sterile water (untreated control) (**A** and **D**) or with 4 mg/mL HICA for 60 min (**B** and **E**), and 120 min (**C** and **F**).

## Transmission electron microscopy (TEM)

Ultrastructural alterations in *B. cereus* NZRM5 and *P. aeruginosa* ATCC25668 cells upon 4 mg/mL HICA treatment were investigated using TEM. Control/ untreated *B. cereus* cells were rod shaped, intact, and presented a complete cell membrane and the cell wall (Fig 6A). After HICA treatment (4 mg/mL HICA for 60 min), several perimortem changes were observed in *B. cereus* cells as shown in Fig 6B. The breakage of the cell wall and cytoplasmic membrane followed by the leakage of cell contents were observed. Lysed cells ('ghost cells") devoid of cytoplasm were also observed.

Significant perimortem changes were also observed in *P. aeruginosa* cells following 120 min incubation with HICA (4 mg/mL). Untreated cells presented an undamaged wavy cell

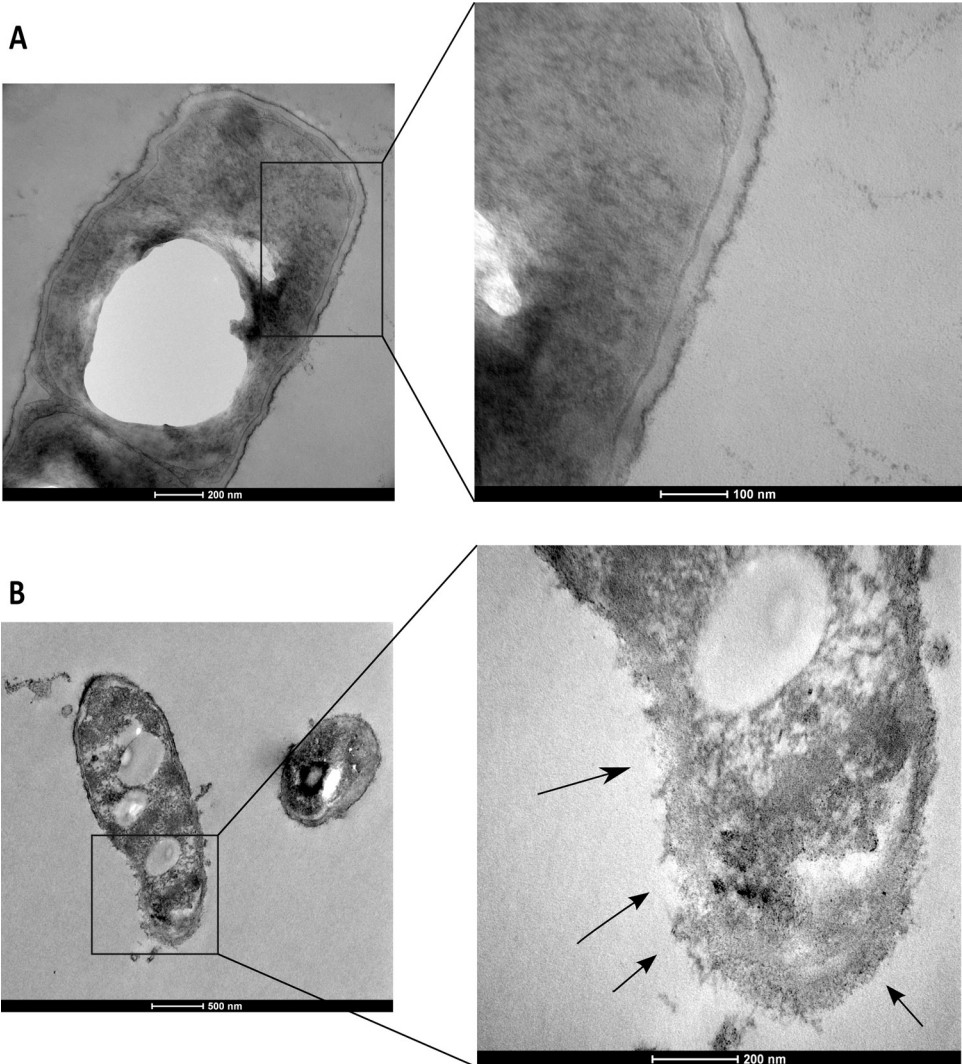

**Fig 6. Transmission electron microscopic micrographs depicting the effect of HICA on the cell morphology of *Bacillus cereus* NZRM5.** The cells were treated with sterile water (untreated control) (**A**) or with 4 mg/mL HICA for 60 min (**B**). Higher magnification of the square region in the left image is provided on the right. Arrows show the locations of the breakage of cell envelope and the release of cellular content.

membrane structure (Fig 7A). Compared to untreated cells, the rupture of cell membrane envelopes, and subsequent release of cellular content were observed in HICA treated cells (Fig 7B).

## Discussion

The current trend for healthy foods has resulted changes in the food processing industries. Demands for food products with minimum processing, minimum food additives and free from synthetic additives ("clean label) are increasing [24]. However, to date only a limited number of natural compounds have been approved to be used in food products [1]. Activity of nisin, which is the most widely used natural antimicrobial in food industry, is mostly limited to controlling Gram-positive bacteria, and its activity against Gram-negative bacteria has been low [25]. In this context, HICA, which is a natural microbial metabolite and physiological

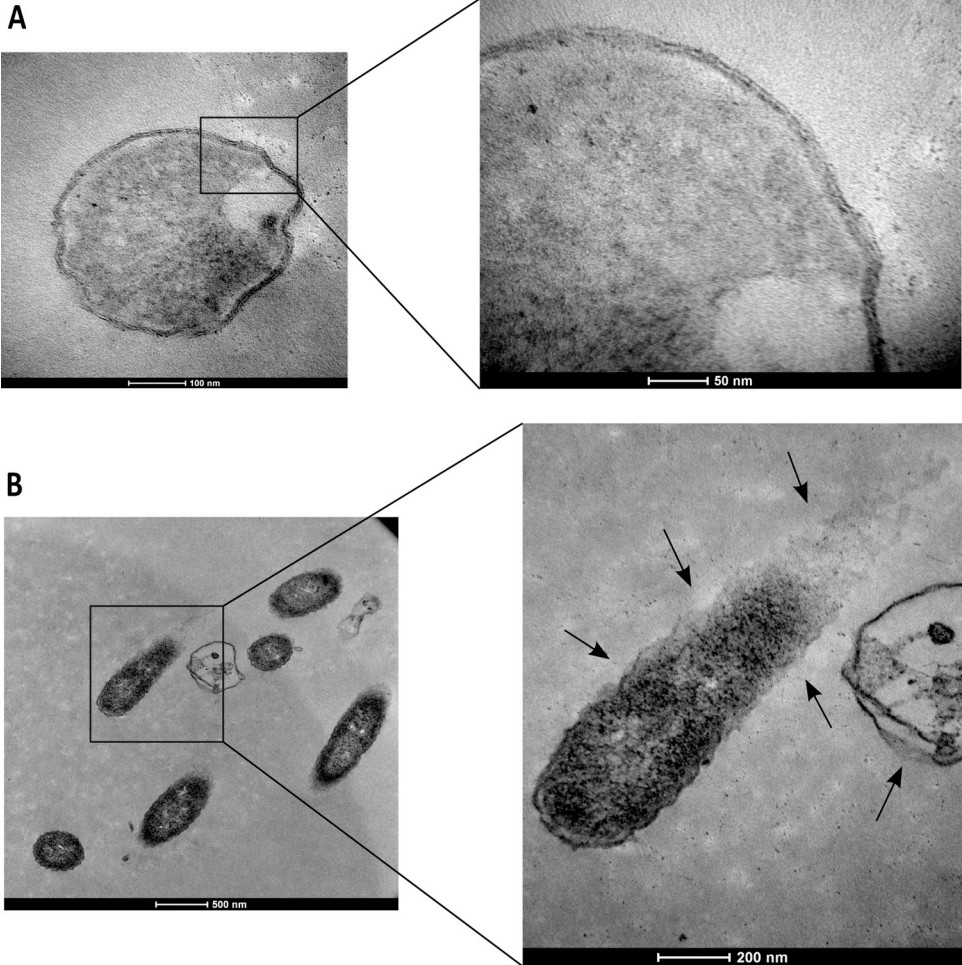

**Fig 7. Transmission electron microscopic micrographs depicting the effect of HICA on the cell morphology of _Pseudomonas aeruginosa_ ATCC25668.** The cells were treated with sterile water (untreated control) (**A**) or with 4 mg/mL HICA for 120 min (**B**). Higher magnification of the square region in the left image is provided on the right. Arrows show the locations of the breakage of cell envelope and the release of cellular content.

agent present in the human body, was investigated for its antibacterial efficacy against different food spoilage and pathogenic bacteria and to provide some understanding of its possible mechanism of action.

The bacterial species used in this study were selected based on their association with food quality and food safety. _B. cereus_, _E. coli_, and _S. aureus_ are three major foodborne pathogens [26]. _Shewanella putrefaciens_, _Serratia proteamaculans_, _Paenibacillus odorifer_, _Bacillus subtilis_, _Bacillus mycoides_, _Pseudomonas lundensis_ and _Pseudomonas fragi_ are known food spoilage bacteria associated with the spoilage of various types of food products such as meat and dairy [27–31]. _P. aeruginosa_ has also been reported as a spoilage bacterium in dairy products [32] and it is a significant opportunistic pathogen causing infections in patients with immunocompromising conditions such as those hospitalised in intensive care units or with cystic fibrosis [33].

Antimicrobial susceptibility testing in this study revealed that HICA was a promising antibacterial agent against all Gram-positive and Gram-negative bacteria tested, with MIC ranging between 0.5–1 mg/mL (Table 1). Both reference strains and environmental isolates of _B. cereus_,

*E. coli*, and *Pseudomonas* species showed the same level of susceptibility to HICA as seen by their MIC values. The effectiveness of HICA against a multidrug-resistant *P. aeruginosa* strain indicated, its potential use against antimicrobial resistant bacteria. A previous study evaluated the cytotoxicity and genotoxicity of HICA and reported that it was safe at concentrations <10 mg/mL [34]. All the MIC values obtained in this study were well below 10 mg/mL. The high MBC values of HICA for *B. cereus* could be attributed to the resistance of *B. cereus* spores present in the growth medium. *B. cereus* spores have been reported to survive under adverse environments such as extreme pH, high temperatures and antimicrobial compounds [35]. Sakko and the team in 2012 investigated the antibacterial efficacy of HICA against clinically significant Gram-positive and Gram-negative human pathogenic bacteria and reported a similar range of MBC values (4.5–36 mg/mL) as reported in the present study (1.0–32 mg/mL) [18].

Further evidence of the antibacterial effect of HICA was provided using the bacterial cell viability assay using three Gram-positive (*B. cereus* NZRM5, *B. cereus* M4, and *S. aureus* NZRM917) and four Gram-negative bacteria (*E. coli* O157:H7 NCTC 1200, *E. coli* AGR3789, *P. aeruginosa* NZRM981, and *P. aeruginosa* NZRM4034). The luminescent signal obtained in the viability assay is proportional to the amount of ATP present in the bacterial culture, which is directly proportional to the number of metabolically active viable cells present in the culture [36]. The results indicated that HICA caused bacterial cell death in a time dependant manner (Fig 1). These results suggest the future application of HICA as an antibacterial agent that may potentially control the growth of both Gram-positive and Gram-negative bacteria responsible for food quality and safety issues. However, this study employed only a few bacterial isolates from each target bacterial species for the assessment of antibacterial activity of HICA. It is well established that there could be strain variations in the same bacterial species to the susceptibility of the same antimicrobial compound. Therefore, HICA needs to be tested against a reasonably large number of strains from each target species to provide concrete evidence on their activity against each target bacterial species. A limitation of this method was not providing the actual viable cell numbers in CFU/mL as the viability was calculated as the percentage of viable cells in the control group.

The cell membrane integrity is important for cell viability and its interruption can lead to metabolic dysfunction and ultimately cell death [37]. The damaging of cell membrane integrity has been recognised as a potential mechanism for cell death [38]. HICA disrupted the cell membrane integrity of three Gram-positive (*B. cereus* NZRM5, *B. cereus* M4, and *S. aureus* NZRM917) and four Gram-negative bacteria (*E. coli* O157:H7 NCTC 1200, *E. coli* AGR3789, *P. aeruginosa* NZRM981, and *P. aeruginosa* NZRM4034) (Fig 2). The loss of cell membrane integrity would lead to the release of ATP from bacterial cells lowering the cell viability as shown in the cell viability assay. Consistent with the cell viability results, tested Gram-positive bacteria showed a more rapid loss of cell membrane integrity than tested Gram-negative bacteria in this study. This could be attributed to the additional outer membrane of Gram-negative bacteria providing an extra barrier for the access of HICA to the cytoplasmic membrane and to the inner cell structures. The outer membrane of Gram-negative bacteria provides an extra layer of protection against antimicrobials by lowering or not allowing access to inner cellular targets such as the cytoplasmic membrane and other intracellular structures [39].

The impact of HICA on the outer membrane permeability of four Gram-negative bacteria (*E. coli* O157:H7 NCTC 1200, *E. coli* AGR3789, *P. aeruginosa* NZRM981, and *P. aeruginosa* NZRM4034) was also investigated using the fluorescence probe 1-*N*-phenylnapthylamine (NPN). NPN is a hydrophobic fluorophore, which cannot effectively penetrate through the outer membranes of Gram-negative bacteria. It gives a weak fluorescence signal in aqueous solution and a strong one when it binds to a phospholipid layer [40]. This assay is well established and has been used elsewhere to describe the outer cell membrane permeabilising

property of antimicrobial molecules [21]. The results in the present study demonstrated that HICA disrupted the integrity of the outer membrane resulting in a loss of barrier function in tested *E. coli* and *P. aeruginosa* strains (Fig 3). This allows HICA to interact with inner cellular targets including the cytoplasmic membrane of Gram-negative bacteria. A concentration range was used in this assay to evaluate the permeabilising capacity of HICA at sub and supra-MIC. The ability of HICA to permeabilise the outer membrane, even at concentrations lower than its MIC, makes it a permeabiliser lacking inherent toxicity at the particular concentration but could sensitise bacteria to other antimicrobial agents when it is used together with other antimicrobial interventions.

The cytoplasmic membrane potential plays a vital role in the chemical and mechanical integrity of bacterial cells and electrical potential [41]. The disruption of the membrane potential of bacterial cells could be either the primary antimicrobial mechanism or contribute to the effectiveness of the compound by allowing access to additional inner molecular targets. The effect of HICA on the cytoplasmic membrane potential of three Gram-positive (*B. cereus* NZRM5, *B. cereus* M4, and *S. aureus* NZRM917) and four Gram-negative bacteria (*E. coli* O157:H7 NCTC 1200, *E. coli* AGR3789, *P. aeruginosa* NZRM981, and *P. aeruginosa* NZRM4034) were examined using the membrane potential-sensitive dye 3,3'-Dipropylthiadi-carbocyanine iodide [DiSC$_3$(5)]. When DiSC$_3$(5) is added to the cell suspension, it accumulates in energised/polarised cells and this movement of the dye from the media results in quenching of the overall fluorescence in the cell suspension. When the cytoplasmic membrane is depolarised, the dye will be rapidly released back into the medium (dequenching), that can be measured fluorometrically as a rapid increase in fluorescence [22]. The fluorescence intensity difference between full energised and depolarised cells are dependent on the bacteria, bacterial cell density, and dye concentration [22]. The current study used a suitable dye concentration and cell density for each bacterium, that clearly showed the effect of HICA on the cytoplasmic membrane of all test bacteria. The results demonstrated that HICA altered the cytoplasmic membrane potential of both Gram-positive and Gram-negative bacteria tested in this study (Fig 4). The cell membrane depolarisation caused by HICA might lead to a change in the barrier properties of the cells and other cell functions leading to cell death. Previous studies have suggested that cytoplasmic membrane depolarisation could result either through the formation of ion-conducting membrane pores, increasing membrane ion permeability or acting as an ion carrier [42, 43].

The examination of bacteria treated with an antimicrobial agent for alterations to their morphology and ultrastructure is commonly used to investigate the mechanism of a potential antimicrobial. It is used as both an early investigative step and confirmatory step for a suspected mechanism [44]. In the present study, *B. cereus* NZRM5 and *P. aeruginosa* ATCC25668 were selected as model Gram-positive and Gram-negative microorganisms, respectively to investigate the morphological changes in bacteria after HICA treatment. Both SEM and TEM images of *B. cereus* demonstrated significant morphological alterations including signs of damage to the cell envelope (roughening and rupturing) and the leak of the cellular content after the HICA treatment (Figs 5–7). The cellular damage caused by HICA in *P. aeruginosa* cells was not very clear in SEM images, whereas TEM images clearly showed the loss of cell membrane integrity and the leakage of cellular content in *P. aeruginosa* cells after HICA treatment.

The evidence in this study indicates that the cell membrane (permeability of outer membrane, depolarisation of cytoplasmic membrane and loss of cell membrane integrity) represents the primary target of HICA. It might target other cellular structures or processes and its activity could be attributed to more than one mechanism. However, further studies are required to understand these mechanisms and other cellular interference of HICA, that may lead to the cell death. A significant feature of HICA, which will enhance its potential

application, is its activity against both Gram-positive and Gram-negative bacteria showing similar MIC values. The low permeability of the outer membrane of Gram-negative bacteria has been reported to prevent the effect of some antimicrobial compounds, which are active against Gram-positive bacteria [45]. This study demonstrated that HICA can disrupt the integrity of the outer membrane of Gram-negative bacteria resulting in the loss of barrier function. This is interesting to note, as HICA is a hydrophobic compound and Gram-negative bacteria have been reported to be more tolerant to hydrophobic antimicrobial compounds compared to Gram-positive bacteria [46].

Higher HICA concentrations reduced the pH of the growth medium (from pH$\approx$ 7.2 to $\approx$ 3 at 32 mg/mL HICA and pH $\approx$ 5.8 at 4mg/ml), suggesting that its antimicrobial activity could be partially or largely associated with the reduced pH inside the cell. It is already documented that low pH can contribute to the antimicrobial effect [47]. For instance, antimicrobial activity of lactic acid has been attributed to its ability to reduce intracellular pH and disrupts cell membrane and affect other cellular processes and structures including enzyme activities, protein, and DNA structure [48, 49]. However, some bacteria such as *E. coli*, and *Bacillus* species possess intrinsic mechanisms to resist low pH and are able to survive in acidic environment [50–53]. In the present study, the pH of the growth medium ($\approx$ 7.2) didn't change with the addition of 0.5 mg/mL HICA (pH $\approx$ 7.0), but changed slightly, from neutral to slight acidic (pH $\approx$ 6.5) with 1 mg/mL of HICA. Both of these concentrations are reported as minimum inhibitory concentrations in the present study. This shows that HICA's antimicrobial effect is not only due to the pH change but other prevailing mechanism/s as well. The current study does not provide information on the antibacterial effect contributed by low pH of HICA at higher concentrations. Therefore, further studies need to be conducted with relevant pH controls to understand the contribution of low pH of HICA on its antimicrobial properties. Nevertheless, the current study revealed HICA's ability to cause bacterial cell death via depolarisation, permeabilisation, rupture of bacterial cell membranes, and subsequent leakage of cellular content.

## Conclusions

This study explored the antibacterial efficacy of HICA against some Gram-positive and Gram-negative bacteria associated with food quality and food safety concerns. The results reveal that HICA is effective in controlling the growth of some Gram-positive and Gram-negative bacteria including a multi-drug resistant *P. aeruginosa* strain, tested in this study. Further studies were conducted to provide some insight into the possible antibacterial mechanism of HICA. HICA exhibited its activity via penetration of the bacterial cell membranes causing depolarisation, permeabilisation, rupture of membranes, subsequent leakage of cellular contents and cell death. Antibacterial activity combined with its previously reported antifungal activity and safety profile suggests that HICA could be considered as a potential natural antimicrobial agent against food spoilage bacteria and pathogens.

## Supporting information

**S1 Table. Bacterial growth inhibition by different HICA concentrations.**
(DOCX)

## Acknowledgments

The authors acknowledge Manawatu Microscopy and Imaging Centre (MMIC) at Massey University, New Zealand for their technical support for SEM and TEM imaging.

## Author Contributions

**Conceptualization:** Amila S. N. W. Pahalagedara, Tanushree B. Gupta.

**Formal analysis:** Amila S. N. W. Pahalagedara.

**Funding acquisition:** Gale Brightwell, Tanushree B. Gupta.

**Investigation:** Amila S. N. W. Pahalagedara, Tanushree B. Gupta.

**Methodology:** Amila S. N. W. Pahalagedara.

**Project administration:** Tanushree B. Gupta.

**Resources:** Tanushree B. Gupta.

**Software:** Amila S. N. W. Pahalagedara.

**Supervision:** Steve Flint, Jon Palmer, Tanushree B. Gupta.

**Validation:** Amila S. N. W. Pahalagedara.

**Writing – original draft:** Amila S. N. W. Pahalagedara.

**Writing – review & editing:** Steve Flint, Jon Palmer, Gale Brightwell, Tanushree B. Gupta.

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
