## [Decision Letter · Decision Letter 0]

6 Dec 2021

PONE-D-21-34303Antimicrobial efficacy and possible mechanism of action of 2-hydroxyisocaproic acid (HICA)PLOS ONE

Dear Dr. Gupta,

Thank you for submitting your manuscript to PLOS ONE. After careful consideration, we feel that it has merit but does not fully meet PLOS ONE’s publication criteria as it currently stands. Therefore, we invite you to submit a revised version of the manuscript that addresses the points raised during the review process. Please, carefully consider the reviewer's comments. Particularly regarding on what you can discuss based on your results. 

We look forward to receiving your revised manuscript.

Kind regards,

Guadalupe Virginia Nevárez-Moorillón, Ph.D.

Academic Editor

PLOS ONE

Journal Requirements:

"Funding

This work was supported by Strategic Science Investment Fund (SSIF), AgResearch Ltd., New Zealand."

Reviewers' comments:

Reviewer's Responses to Questions

5. Review Comments to the Author

Reviewer #1: This is an interesting paper with potentially significant findings.

However, there are multiple important points that require clarification:

1. Which HICA was used? Was it DL-HICA?

2. What was the pH of HICA solution and the final pH during the incubations?

HICA comes in two isoforms of which DL is the more active one. It only stays in DL isoform in low pH and in more neutral conditions it becomes a racemic solution.

Please clarify in the methods and discuss in the discussion if relevant.

There is no mention of what controls were used. Looking at Figure 1 it looks like there was a non-HICA containing control.

1. What was the pH of this non-HICA containing control?

2. What other controls were used in the various experiments?

3. Please clarify for all experiments what controls were used.

The Discussion is far too long and starts off as a second introduction. Please focus on discussing your own results here. Also, please discuss the limitations of the study and the methods used. Please do not speculate beyond your results.

Reviewer #2: This study evaluates antibacterial effect of a natural compound 2-hydroxyisocaproic acid and shows new interesting perspectives of possible antibacterial mode of action this agent. The mode of action has been studied with several different methods. There are several points in the study, which should be considered, and corrected or explained.

1. Title and the text: Is it better to use antibacterial term together with or instead of antimicrobial, because it could be more precise in this context?

2. Materials and methods: Row 107 "our laboratory culture collection." Which is the name of the organization?

3. Materials and methods: The testings of antimicrobial susceptibility of bacteria are told to be performed with three replicates, but obviously not repeated in this study. Therefore, the degree of evidence should be considered. Antibacterial efficacy of 2-hydroxyisocaproic acid has been reported earlier against Escherichia coli, Pseudomonas aerunginosa and Staphylococcus aureus, but there are several bacterial species included in this study, which are tested for the first time. Is the evidence of their susceptibility for HICA sufficient? Repetition of the efficacy testings of bacterial species is recommended.

4. Materials and methods: What is the pH-value, in which the experiments have been performed? This might have an influence on antibacterial effect.

5. Materials and methods: Were the final test concentrations, for which the bacterial strains and isolates were exposed 0,5-32mg/mL or lower after adding 50 microliter of bacterial suspension and 50 microliter of MHB to 100 microliter of test solutions? This should be explained more clearly in the text, and corrected to the results, if needed.

6. Materials and methods: In the row 125: "or water (untreated control)". Isn't this sterile water?

7. Materials and methods: Sterile water in MHB acted as a negative control in the efficacy testings. Were there used any positive control, which is known to kill the bacterial strains?

8. Materials and methods: There are multiple methods used in this study, but all bacterial strains, which are introduced in the beginning of Materials and method section are not tested in all experiments. The bacterial strains and isolates, which are tested in different experiments (range 2-14) should be clarified in the text. Because of using the terms Gram-positive and negative bacteria readers may suppose that all 14 strains and isolates have been tested in all experiments.

9. Statistical analyses in the bacterial susceptibility testings are told to been done with single factor ANOVA, for which the results should be normally distributed. Three replicates per group is also the lowest number of replicates for these analyses. Is the normal distribution of the results within the groups confirmed or has this been possible?

10. Results: Row 294: "Florescence" should be corrected fluorescence.

11. Results: In the row 329 there is written: "HICA addition induced a rapid depolarisation of the cytoplasmic membrane of both Gram-positive and Gram-negative bacteria as evident by rapid increase in fluorescence intensities." However, the colors blue/red possibly shows the opposite in the Figure. This should be corrected.

12. Discussion: In the Discussion there is a risk for misinterpretation of the results. For instance, the term of Gram-positive bacteria as plural may end up to larger adaptation of the results than is meaningful. As earlier was mentioned, all bacterial strains (n=14) included in the first part of the study are not tested in the other parts of the study. The number of bacterial strains tested in them varied from 2 to 7. So, the results should be adapted only on the tested bacterial strains and isolates of each experiment, not on all Gram-positive and Gram-negative bacteria in general. For clarity, the names of tested bacterial strains and their designation codes should be written in the Discussion text. Some examples are below.

a. "Further evidence for the antimicrobial effect of HICA was provided using the bacterial cell

viability assay of several Gram-positive and Gram-negative bacteria." Comment: The exact amount of Gram-positive and Gram-negative bacterial strains should be mentioned, like three Gram-positive bacterial and four Gram-negative bacterial strains. And the bacterial strains tested should be mentioned by their names and designation codes (or by other distinctive manners).

b. "HICA disrupted the cell membrane integrity of both Gram-positive and Gram-negative bacteria (Fig 2)." Comment: Please, tell also the exact amount of tested bacterial strains (n=7?) and individualize them in this part of the study.

c. "This characteristic of NPN is used to examine the permeability of the outer membrane of Gram-negative bacteria [21]." Comment: Name exactly the bacterial strains, which were tested in this method of the study (n=4?).

d. "The effect of HICA on the cytoplasmic membrane potential of both Gram-positive and Gram-negative bacteria were examined using the membrane potential-sensitive dye 3,3'-Dipropylthiadicarbocyanine iodide [DiSC3(5)]." Comment: Please, individualize the bacterial strains, which were tested in this part of the study (n=2) for the reason mentioned earlier.

e. "In the present study, B. cereus and P. aeruginosa were selected as model microorganisms to investigate the morphological changes in Gram-positive and Gram-negative bacteria after HICA treatment." Comment: Please, individualize the exact bacterial strains, which were used in both electron microscopy methods, because there were some other strains from these species included in the study too.

13. Discussion: "sub and supra-MIC" Comment: Is this expressed in correct language? Should it be written as: below and beyond, or under and over the minimum inhibitory concentrations?

---

## [Author Response · Author response to Decision Letter 0]

18 Jan 2022

We thank reviewers for reviewing our work.

Reviewer 1

1. Which HICA was used? Was it DL-HICA? 

Yes, DL-HICA was used in this study. That information was added in the text under materials and methods (L 118-120) 

2. What was the pH of HICA solution and the final pH during the incubations?

HICA comes in two isoforms of which DL is the more active one. It only stays in DL isoform in low pH and in more neutral conditions it becomes a racemic solution.

Please clarify in the methods and discuss in the discussion if relevant. 

Information about HICA isoform used in this study and the pH during the incubation was added in the text (L 118-120) and (L 136, L 151, L546, L548), respectively. DL-HICA was used in our study and other studies elsewhere that reported antimicrobial activity. But there is no information in the literature about the relationship between HICA’s antimicrobial activity and its isoform. So, we have not further commented on it. 

3. There is no mention of what controls were used. Looking at Figure 1 it looks like there was a non-HICA containing control.

1. What was the pH of this non-HICA containing control? 

2. What other controls were used in the various experiments? 

3. Please clarify for all experiments what controls were used.

Non-HICA control used was growth medium + cells + sterile water (pH ≈ 7.2), included in the text (L546)

All the experiments were performed with relevant untreated controls and the HICA-treated groups were compared against respective untreated controls. This information was included in all relevant methodologies.

4. The Discussion is far too long and starts off as a second introduction. Please focus on discussing your own results here. Also, please discuss the limitations of the study and the methods used. Please do not speculate beyond your results

Reduced the amount of introductory and methodology information in the discussion section and focused more on discussing results and limitations. 

Reviewer 2

1. Title and the text: Is it better to use antibacterial term together with or instead of antimicrobial, because it could be more precise in this context?

Changed the term ‘antimicrobial’ to ‘antibacterial’ in the title and the other places in the text as suggested

2. Materials and methods: Row 107 "our laboratory culture collection." Which is the name of the organization?

Included in the text - Food System Integrity Team, AgResearch Ltd., New Zealand (L105-106)

3. Materials and methods: The testing of antimicrobial susceptibility of bacteria are told to be performed with three replicates, but obviously not repeated in this study. 

Therefore, the degree of evidence should be considered. Antibacterial efficacy of 2-hydroxyisocaproic acid has been reported earlier against Escherichia coli, Pseudomonas aeruginosa and Staphylococcus aureus, but there are several bacterial species included in this study, which are tested for the first time. Is the evidence of their susceptibility for HICA sufficient? Repetition of the efficacy testings of bacterial species is recommended.

The MIC and MBC values in the Table 1 were obtained by conducting MIC and MBC assays with three replicates of HICA with a dilution series consisting final concentrations, 32 mg/mL, 16 mg/mL, 8 mg/mL, 4 mg/mL, 2 mg/mL, 1 mg/mL, 0.5 mg/mL and 0.25 mg/mL. The results of all three replicates were consistent showing the same minimum inhibitory concentration and minimum bactericidal concentrations. For instance, all three replicates of HICA showed that ≥ 0.5 mg/mL HICA concentrations could inhibit the growth of Shewanella putrefaciens SM26 completely in the MHB (no OD change) and ≥ 1 mg/mL HICA concentrations resulted no colonies on SBA plates. Therefore, minimum inhibitory concentration (MIC) and minimum bactericidal concentration (MBC) of HICA against Shewanella putrefaciens SM26 were 0.5 mg/mL and 1 mg/mL respectively. Now, the use of three replicates of HICA to assess the MIC and MBC against all test bacteria is included in the method section (L141-143 and L258). Supplementary file_S1(Table S1). 

4. Materials and methods: What is the pH-value, in which the experiments have been performed? This might have an influence on antibacterial effect.

Different pH values have been documented in the text, including discussion, (L 136, L 151, L546, L548) also, (L544 – L550). 

5. Materials and methods: Were the final test concentrations, for which the bacterial strains and isolates were exposed 0,5-32mg/mL or lower after adding 50 microliter of bacterial suspension and 50 microliter of MHB to 100 microliter of test solutions? This should be explained more clearly in the text, and corrected to the results, if needed.

Included the working concentrations of HICA in the text to clarify the final HICA concentrations after two-fold dilution in the assay with medium and the cells. (L127-132).

6. Materials and methods: In the row 125: "or water (untreated control)". Isn't this sterile water?

Yes, it is sterile water. Included in the text (L132)

7. Materials and methods: Sterile water in MHB acted as a negative control in the efficacy testing. Were there used any positive control, which is known to kill the bacterial strains?

No positive control was used in the efficacy assay as there was no intention of comparing the antimicrobial efficacy of HICA to any known compound. Purpose was to determine the MIC and MBC values.

8. Materials and methods: There are multiple methods used in this study, but all bacterial strains, which are introduced in the beginning of Materials and method section are not tested in all experiments. The bacterial strains and isolates, which are tested in different experiments (range 2-14) should be clarified in the text. Because of using the terms Gram-positive and negative bacteria readers may suppose that all 14 strains and isolates have been tested in all experiments.

Included bacteria names in the methodologies 

9. Statistical analyses in the bacterial susceptibility testings are told to been done with single factor ANOVA, for which the results should be normally distributed. Three replicates per group is also the lowest number of replicates for these analyses. Is the normal distribution of the results within the groups confirmed or has this been possible?

Three replicates have been employed and accepted as the minimum number of replicates to determine the statistical significance between groups using one-way ANOVA [1]. Normal distribution of the data within the groups was confirmed by Shapiro-Wilk test of normality before performing ANOVA.

10. Results: Row 294: "Florescence" should be corrected fluorescence.

Corrected (L315)

11. Results: In the row 329 there is written: "HICA addition induced a rapid depolarisation of the cytoplasmic membrane of both Gram-positive and Gram-negative bacteria as evident by rapid increase in fluorescence intensities." However, the colors blue/red possibly shows the opposite in the Figure. This should be corrected.

Corrected the colors in the figure 4 caption (L359-360)

12. Discussion: In the Discussion there is a risk for misinterpretation of the results. For instance, the term of Gram-positive bacteria as plural may end up to larger adaptation of the results than is meaningful. As earlier was mentioned, all bacterial strains (n=14) included in the first part of the study are not tested in the other parts of the study. The number of bacterial strains tested in them varied from 2 to 7. So, the results should be adapted only on the tested bacterial strains and isolates of each experiment, not on all Gram-positive and Gram-negative bacteria in general. 

For clarity, the names of tested bacterial strains and their designation codes should be written in the discussion text. Some examples are below.

a. "Further evidence for the antimicrobial effect of HICA was provided using the bacterial cell

viability assay of several Gram-positive and Gram-negative bacteria." Comment: The exact amount of Gram-positive and Gram-negative bacterial strains should be mentioned, like three Gram-positive bacterial and four Gram-negative bacterial strains. And the bacterial strains tested should be mentioned by their names and designation codes (or by other distinctive manners). 

Included the number of bacteria and their names in the text (L450-L453)

b. "HICA disrupted the cell membrane integrity of both Gram-positive and Gram-negative bacteria (Fig 2)." Comment: Please, tell also the exact amount of tested bacterial strains (n=7?) and individualize them in this part of the study.

Included the number of bacteria and their names in the text (L467-472)

c. "This characteristic of NPN is used to examine the permeability of the outer membrane of Gram-negative bacteria [21]." Comment: Name exactly the bacterial strains, which were tested in this method of the study (n=4?).

Included the number of Gram-negative bacteria and their names in the text (L482-483)

d. "The effect of HICA on the cytoplasmic membrane potential of both Gram-positive and Gram-negative bacteria were examined using the membrane potential-sensitive dye 3,3'-Dipropylthiadicarbocyanine iodide [DiSC3(5)]." Comment: Please, individualize the bacterial strains, which were tested in this part of the study (n=2) for the reason mentioned earlier.

Included the number of bacteria and their names in the text (L502-505)

e. "In the present study, B. cereus and P. aeruginosa were selected as model microorganisms to investigate the morphological changes in Gram-positive and Gram-negative bacteria after HICA treatment." Comment: Please, individualize the exact bacterial strains, which were used in both electron microscopy methods, because there were some other strains from these species included in the study too.

Included strain names in the text (L524)

13. Discussion: "sub and supra-MIC" Comment: Is this expressed in correct language? Should it be written as: below and beyond, or under and over the minimum inhibitory concentrations?

The terms “sub-MIC” and “supra-MIC” to describe lower and higher concentrations of MIC of a particular antimicrobial compound are used in previous antimicrobial research [2-4].

References

[1] Liu X, Zhang M, Meng X, He X, Zhao W, Liu Y, et al. Inactivation and membrane damage mechanism of slightly acidic electrolyzed water on Pseudomonas deceptionensis CM2. Molecules (Basel, Switzerland). 2021;26:1012.

[2] Juma A, Lemoine P, Simpson ABJ, Murray J, O’Hagan BMG, Naughton PJ, et al. Microscopic investigation of the combined use of antibiotics and biosurfactants on methicillin resistant Staphylococcus aureus. Frontiers in Microbiology. 2020;11.

[3] Odenholt I, Holm SE, Cars O. Effects of supra- and sub-MIC benzylpenicillin concentrations on group A β-haemolytic streptococci during the postantibiotic phase in vivo. Journal of Antimicrobial Chemotherapy. 1990;26:193-201.

[4] Odenholt-Tornqvist I, Löwdin E, Cars O. Pharmacodynamic effects of subinhibitory concentrations of beta-lactam antibiotics in vitro. Antimicrobial Agents and Chemotherapy. 1991;35:1834-9.

---

## [Decision Letter · Decision Letter 1]

17 Feb 2022

PONE-D-21-34303R1Antibacterial efficacy and possible mechanism of action of 2-hydroxyisocaproic acid (HICA)PLOS ONE

Dear Dr. Gupta,

Thank you for submitting your manuscript to PLOS ONE. After careful consideration, we feel that it has merit but does not fully meet PLOS ONE’s publication criteria as it currently stands. Therefore, we invite you to submit a revised version of the manuscript that addresses the points raised during the review process.

Please consider mainly the first suggestion of the reviewer, that I consider will improve the document.  Please submit your revised manuscript by Apr 03 2022 11:59PM. If you will need more time than this to complete your revisions, please reply to this message or contact the journal office at plosone@plos.org. Please include the following items when submitting your revised manuscript:A rebuttal letter that responds to each point raised by the academic editor and reviewer(s). You should upload this letter as a separate file labeled 'Response to Reviewers'.A marked-up copy of your manuscript that highlights changes made to the original version. You should upload this as a separate file labeled 'Revised Manuscript with Track Changes'.An unmarked version of your revised paper without tracked changes. You should upload this as a separate file labeled 'Manuscript'.If applicable, we recommend that you deposit your laboratory protocols in protocols.io to enhance the reproducibility of your results. Protocols.io assigns your protocol its own identifier (DOI) so that it can be cited independently in the future. For instructions see: https://journals.plos.org/plosone/s/submission-guidelines#loc-laboratory-protocols. Additionally, PLOS ONE offers an option for publishing peer-reviewed Lab Protocol articles, which describe protocols hosted on protocols.io. Read more information on sharing protocols at https://plos.org/protocols?utm_medium=editorial-email&utm_source=authorletters&utm_campaign=protocols.

We look forward to receiving your revised manuscript.

Kind regards,

Guadalupe Virginia Nevárez-Moorillón, Ph.D.

Academic Editor

PLOS ONE

Journal Requirements:

Reviewers' comments:

Reviewer's Responses to Questions

6. Review Comments to the Author

Reviewer #1: The manuscript has improved but there are still a number of outstanding issues:

1. The lack of pH controls needs to be at least discussed. Ideally the experiments would be repeated with them but an insightful discussion may suffice. It is well known that some microbes tolerate low pH poorly whereas others benefit from it. It would be important to know if the effects seen are actually due to the antimicrobial effect of HICA or the low pH (as low as 3 as stated in the methods).

2.I note that the word antimicrobial has been changed to antibacterial throughout. I presume this is due to comments made by the reviewers. The change is correct in some instances but in others not. HICA does have antifungal activity (see reference below) and when described more broadly antimicrobial activity is more appropriate than antibacterial activity. The authors need to go through each of those changes and check what exactly they mean and which of the two words is more appropriate.

Sakko M, Moore C, Novak-Frazer L, Rautemaa V, Sorsa T, Hietala P, Järvinen A, Bowyer P, Tjäderhane L, Rautemaa R.2-hydroxyisocaproic acid is fungicidal for Candida and Aspergillus species. Mycoses. 2014 Apr;57(4):214-21. doi: 10.1111/myc.12145. Epub 2013 Oct 11.

Nieminen MT, Novak-Frazer L, Rautemaa V, Rajendran R, Sorsa T, Ramage G, Bowyer P, Rautemaa R. A novel antifungal is active against Candida albicans biofilms and inhibits mutagenic acetaldehyde production in vitro. PLoS One. 2014 May 27;9(5):e97864. doi: 10.1371/journal.pone.0097864.

A recent publication on HICA activity against anaerobic bacteria is also relevant for this paper.

Sakko M, Rautemaa-Richardson R, Sakko S, Richardson M, Sorsa T. Antibacterial Activity of 2-Hydroxyisocaproic Acid (HICA) Against Obligate Anaerobic Bacterial Species Associated With Periodontal Disease. Microbiol Insights. 2021 Oct 21

---

## [Author Response · Author response to Decision Letter 1]

16 Mar 2022

We thank reviewers for reviewing our work.

1. The lack of pH controls needs to be at least discussed. Ideally the experiments would be repeated with them, but an insightful discussion may suffice. It is well known that some microbes tolerate low pH poorly whereas others benefit from it. It would be important to know if the effects seen are actually due to the antimicrobial effect of HICA or the low pH (as low as 3 as stated in the methods).

This has been identified and discussed in the discussion segment. The lack of information on clarifying the relationship between low pH of HICA and its antimicrobial activities in the current work and the need for future studies to fill that knowledge have been identified and discussed. Please see lines 547 to 562

2. I note that the word antimicrobial has been changed to antibacterial throughout. I presume this is due to comments made by the reviewers. The change is correct in some instances but in others not. HICA does have antifungal activity (see reference below) and when described more broadly antimicrobial activity is more appropriate than antibacterial activity. The authors need to go through each of those changes and check what exactly they mean and which of the two words is more appropriate.

We thoroughly checked the manuscript and changed the terminology accordingly.

---

## [Editor Report · Decision Letter 2]

21 Mar 2022

Antibacterial efficacy and possible mechanism of action of 2-hydroxyisocaproic acid (HICA)

PONE-D-21-34303R2

Dear Dr. Gupta,

We’re pleased to inform you that your manuscript has been judged scientifically suitable for publication and will be formally accepted for publication once it meets all outstanding technical requirements.

Kind regards,

Guadalupe Virginia Nevárez-Moorillón, Ph.D.

Academic Editor

PLOS ONE
---

## [Editor Report · Acceptance letter]

24 Mar 2022

PONE-D-21-34303R2 

Antibacterial efficacy and possible mechanism of action of 2-hydroxyisocaproic acid (HICA) 

Dear Dr. Gupta:

I'm pleased to inform you that your manuscript has been deemed suitable for publication in PLOS ONE. Congratulations! Your manuscript is now with our production department. 

Kind regards, 

on behalf of

Dr. Guadalupe Virginia Nevárez-Moorillón 

Academic Editor

PLOS ONE